# Hybrid Systems of Nanofibers and Polymeric Nanoparticles for Biological Application and Delivery Systems

**DOI:** 10.3390/mi14010208

**Published:** 2023-01-14

**Authors:** Hever Yuritzy Vargas-Molinero, Aracely Serrano-Medina, Kenia Palomino-Vizcaino, Eduardo Alberto López-Maldonado, Luis Jesús Villarreal-Gómez, Graciela Lizeth Pérez-González, José Manuel Cornejo-Bravo

**Affiliations:** 1Facultad de Ciencias Químicas e Ingeniería, Universidad Autónoma de Baja California, Tijuana 22390, Mexico; 2Facultad de Medicina y Psicología, Universidad Autónoma de Baja California, Tijuana 22390, Mexico; 3Facultad de Ciencias de la Ingeniería y Tecnología, Universidad Autónoma de Baja California, Tijuana 22427, Mexico

**Keywords:** drug delivery, nanoparticles, nanofibers, hybrid system, nanomedicine, electrospinning

## Abstract

Nanomedicine is a new discipline resulting from the combination of nanotechnology and biomedicine. Nanomedicine has contributed to the development of new and improved treatments, diagnoses, and therapies. In this field, nanoparticles have notable importance due to their unique properties and characteristics, which are useful in different applications, including tissue engineering, biomarkers, and drug delivery systems. Electrospinning is a versatile technique used to produce fibrous mats. The high surface area of the electrospun mats makes them suitable for applications in fields using nanoparticles. Electrospun mats are used for tissue engineering, wound dressing, water-treatment filters, biosensors, nanocomposites, medical implants, protective clothing materials, cosmetics, and drug delivery systems. The combination of nanoparticles with nanofibers creates hybrid systems that acquire properties that differ from their components’ characteristics. By utilizing nanoparticles and nanofibers composed of dissimilar polymers, the two synergize to improve the overall performance of electrospinning mats and nanoparticles. This review summarizes the hybrid systems of polymeric nanoparticles and polymeric nanofibers, critically analyzing how the combination improves the properties of the materials and contributes to the reduction of some disadvantages found in nanometric devices and systems.

## 1. Introduction

Nanoparticles have been essential as an innovative technology in drug delivery systems. With chemotherapeutic drugs, for example, greater specificity is needed to reduce adverse reactions during cancer treatments. These nanometric systems are able to deliver more concentrated doses in close proximity to tumor tissue due to the effects of improved permeability and retention. This, in turn, reduces the exposure of the drug to healthy tissues, and limits the distribution of chemotherapeutics to the targeted tumor tissue. This results in a more precise and less toxic dosage during treatment. Nanoparticles can also be used as carriers to penetrate the blood-brain barrier (BBB) and deliver antineoplastic drugs to treat brain tumors [1]. Another property offered by nanoparticles is to improve the bioavailability of bioactive molecules prone to enzymatic and hydrolytic degradation. Nanoparticles can be used in this capacity as oral administration vehicles for the encapsulation of peptides and proteins, which protects them from gastrointestinal barriers. They can also be used for gene delivery, improving absorption by increasing the cellular permeability of macromolecules. Nanoparticles can be targeted with specific ligands to provide better strategies for therapeutics [2,3].

Electrospun mats, on the other hand, are being studied for different purposes, such as tissue engineering, wound dressing, water treatment filters, biosensors, nanocomposites, medical implants, protective clothing materials, cosmetics, and drug delivery systems [4]. The nanofiber mats formed by electrospinning have different qualities, such as improving the dissolution of hydrophobic drugs and presenting greater release than other systems (e.g., hydrogels, films). The high surface area to volume ratio and many inter-/intra fibrous pores contribute to drug deposition and accumulation [5].

Independently, nanoparticles and polymeric nanofibers present some limitations and disadvantages. They are prone to degradation when used as implantable devices, affecting the device’s sensitivity and useful life expectancy [6]. Nanoparticles can present difficulties during their physical handling, either in liquid or dry forms, due to their small size and large surface area, which make them prone to aggregation. These properties also limit the drug load to a very small amount [2]. As for nanofibers, they present morphological characteristics that must be adjusted for the desired application. Nanofibers must meet specific requirements regarding their mechanical properties and wettability for use as biomedical membranes. Hydrophobic nanofiber mats have been shown to cause plugging and decreased flow due to air pockets trapped within the mats. The addition of nanoparticles can assist in developing a system with more suitable characteristics by modifying properties such as the tensile strength of the compound, its roughness, and the wettability of the system [7,8].

This review summarizes hybrid systems of polymeric nanoparticles and polymeric nanofibers and critically analyzes how the combination contributes to the reduction of some disadvantages found in systems that only contain one type of nanotechnology [9,10,11,12].

Insightful reviews were found regarding hybrid systems, mainly reporting on fibrous mats containing inorganic nanoparticles [9,13].The current review, however, focuses on polymeric nanoparticles from the preparation of the polymeric nanoparticles to drug loading and the incorporation into the nanofibers, which present particular challenges, but also present versatile applications within biomedicine [14]. This review presents an introduction to the electrodynamic method for the production of nanomaterials as well as an overview of nanoparticle synthesis, drug loading and incorporation of nanofibers. In the following sections, the materials used and applications will be discussed.

## 2. The Electrohydrodynamic Method

The electrodynamic methods consist of electrospinning for the creation of nanofibers and electrospray for the development of nanoparticles. These are similar techniques that take into account some experimental conditions but with different parameters and produce different material shapes, sizes, and characteristics [15]. The electrohydrodynamic method can make different types, kinds, and sizes of particles, mats, and a blend of both structure types [14,16]. The general setup for the electrodynamic method, either to form fibers (Figure 1a) or particles (Figure 1b), consists of applying high voltage to the tip of a needle of a syringe containing a polymer or biopolymer solution to a conductive metallic collector. First, the plunger is pushed by a syringe-pump, which produces the output of the solution at controlled parameters. Through the tip of the syringe, a cone is formed, called “Taylor’s cone”. The collector must be made of a conductive material that serves as a support for the material that is deposited in it, and the collector must be connected to the electric ground of the high-voltage system.

These particles and fibers have qualities that put them at an advantage against other types of materials. One of the most commonly mentioned are the small sizes, which range from micrometers to nanometers and qualifies them as nanotechnology; they also have very desirable characteristics, such as high porosity and small pore size. In addition, they have a high surface area to volume ratio, which means greater exposure to the environment [17]. The products of electrodynamic processes have a huge variety of applications in different areas, such as the food industry and the biomedical industry. Its applications include the encapsulation of substances, filtration, packaging development, and enzyme fixation, among others [18].

Electrospray follows the same principles as electrospinning. A charged solution passes through a needle, leading to the ionization and atomization of the solution, and produces microscopic, charged droplets. The solvent evaporates when nano- and micro-sized droplets are produced and collected on a grounded plate. Electrospray has great encapsulation capacity, especially for hydrophilic particles in hydrophobic carriers [19,20]. Nanoparticles formed by electrospray are suitable for drug-carrying applications. Different morphologies can be obtained in the nanoparticles, such as solid spheres, ellipses, or hollow spheres. In addition to morphology, size can also be controlled [21]. Another use for electrospray is the formation of membranes in conjunction with nanofibers, a useful configuration for filtering technologies [22,23].

### 2.1. Electrospinning

One of the products of the electrodynamic process is nanofibers. The electrospun nanofibers made from polymeric materials can be biocompatible, biodegradable, have a high load capacity, and encapsulation efficiency. Another advantage is that they can be produced cost-effectively. Additional features can be added to these electrospinning mats by using copolymers or functionalizing the solution with bioactive molecules. Coating the nanofiber’s surface with layers of polymeric materials, for example, which have desired physicochemical properties such as pH sensitivity. This can help to control the degradation of the nanofibers at different pH levels and provide stability in the cellular environment [15]. The electrospinning technique is easy to use, simple, unique, and versatile. With it, nanofibers of different polymeric materials can be created; about 100 different electrospun systems have already been reported, including organic and inorganic polymers. With the electrospinning technique, different strategies can be used to form nanostructures with the necessary qualities to be used as targeted delivery systems. These delivery systems can provide advantages such as increasing the therapeutic efficacy and reducing the toxicity of a drug [24]. Electrospinning nanofibers are networks capable of guiding and promoting cell migration. The properties of the nanofibers allow for cell proliferation in cellular microenvironments, which promotes angiogenesis, aids in healing, and reduces the inflammation response [25].

Nanofibers from the electrodynamic method can be used in cancer treatments, especially in solid tumors, where the implantation of these devices aids in the elimination of remaining cancer cells after the removal of a tumor as well as the regeneration of tissue in the cavity resulting from the tumor removal process. Since they have a nanostructure similar to the cellular matrix, the surface morphology can be modified for the desired properties to solve these needs [26].

Through the electrospinning method, hydrophobic and hydrophilic polymers can be mixed to modulate the rate of drug release according to the pharmacotherapeutic requirements. It is well known that prolonged release of hydrophilic drugs is difficult to maintain since they tend to dissolve very quickly in aqueous media. Through electrospinning, a hydrophilic drug can be loaded onto the nanofibers, and release can be controlled with the help of polymers with hydrophobic characteristics. Combined with the nanofibers, they minimize the drug delivery in the initial stage and keep it continuous, avoiding the typical explosive release of hydrophilic drugs (“burst release”). Another electrodynamic method that can be used when this approach does not work is coaxial electrospinning. In this method, a coaxial syringe is used for the mixture of two concentric fluids, which results in coaxial mats, obtaining nanofibers with a core-shell structure [27]. This method is applied to obtain encapsulated drugs within the nanofiber. These nanofibers, called core-sheath types, can be developed precisely. The interior of the nanofiber (core) is composed of the drug and a hydrophilic polymer, while the exterior (shell) consists of a hydrophobic polymer and controls the rate of release [27,28,29].

Electrospinning can also be used to prepare multilayer systems inspired by the treatment of skin wounds. The systems consist of layers with different physicochemical characteristics (Janus systems). The exterior layer is superhydrophobic to avoid bacterial proliferation, while the internal layer is superhydrophilic to release antibiotics and wound healing drugs, with a middle layer that delays exudate loss [30].

Reactive electrospinning is one of the varieties of electrospinning used to obtain nanofiber mats that resemble and share structural characteristics with biological tissue. Modified electrospinning techniques are used to copy or approximate biological qualities, obtaining anisotropy of the fibers, rigidity, mechanical properties, and porosity similar to living tissue. Functionalized joints are created for the interconnection of the fibers by combining electrospinning with a crosslinking process, either during electrospinning or after the formation of the nanofibers [31,32].

Electrospinning can be photoreactive if the fluid jet of the polymer solution is exposed to radiation, UV light, or laser, giving rise to photocrosslinking. When the use of chemical crosslinkers is necessary in the process, it is referred to as chemical reactive electrospinning [31].

Li-Li Wu et al., developed a nanocomposite of polyaniline and polyacrylonitrile that was electrospun with polymeric materials based on L-lysine due to their large number of amino groups and is used as an ammonium sensor. The nanocomposite is prepared by electrospinning a homogeneous solution of poly(L-Lysine) with base polymers to form uniform nanofibers as compared to fibers prepared with nanogels based on L-lysine. The nanocomposite with nanogels showed a better response to ammonia because the nanostructure of the nanogels resulted in a rougher surface that increases the surface area of the membrane [33].

### 2.2. Limitations in the Electrohydrodynamic Processes

The synthetic polymers used in electrospinning meet the necessary specifications to be electrospun, such as forming stable solutions in different solvents or having an adequate molecular weight. In contrast, natural polymers that are more focused on meeting needs in the field of biomedicine have more complications to meet qualities that facilitate the electrospinning process. Natural polymers show difficulties in being electrospun. Among these polymers are chitosan, collagen, gelatin, silk fibroins, and fibrinogen. They present instability in their mechanical qualities, low solubility, and high viscosity. This limits their ability to be electrospun [34,35,36]. The mechanical structure of electrospun fibers is of great importance as it influences the expected performance of devices and systems made from these nanofibers. Since these mechanical properties have an effect, for example, on cell growth and migration [37].

Systems and devices are developed for different purposes as required by different needs and individual characteristics. When the design of the nanofibers is projected to be a drug delivery system, another limitation that arises is the incompatibility of drugs and hydrophilic polymers, where the drug can concentrate on the surface of the nanofibers. The uneven distribution can cause an undesired quick release.

Regarding the mechanical properties, they must not only comply with a series of fixed qualities because they are generally developed in dry atmospheres. It is necessary to adjust and test their mechanical changes in environments similar to where they will be used, (generally humid environments). If they are exposed to fluids, for instance, the mechanical properties must be flexible, soft, and resistant at their working conditions [38,39,40].

## 3. Nanoparticles’ Preparation and Drug Loading

Nanoparticles have great potential for use in many different applications. These include manufacturing, materials, energy harvesting, electronics, the environment, mechanical industries, and medications. The biomedicine research field in particular takes advantage of the different physicochemical properties of the nanoparticles [41].

The structures of these nanoparticles are more complex than a molecule; their composition is organized into three layers: the surface, the shell, and the core. This structure provides the nanoparticles with desirable qualities, such as the combination of different materials and the association of molecules at the surface layer [42,43]. Combining more than one material can provide the nanoparticles with unique properties. Additionally, the variety of blend compositions and formulations make nanoparticles with variable drug delivery capabilities to incorporate hydrophobic and hydrophilic molecules with different molecular weights to control their release [44,45].

Polymeric nanoparticles have been developed with different antineoplastic drugs, such as paclitaxel, docetaxel, cisplatin, 5-fluorouracil, and doxorubicin, among others. Depending on the required needs of the polymeric nanocarrier and the drug to be loaded, the most appropriate type of method for the development of the nanoparticles is chosen. There are several methods to prepare organic nanoparticles. They can be prepared from preformed polymers or by polymerization of monomers [46,47].

As mentioned previously, the electrodynamic method can be used to prepare nanoparticles. However, the most frequently used techniques for the preparation of nanoparticles from polymeric materials are the emulsion-evaporation and emulsion-diffusion techniques [47,48,49,50].

Regarding the synthesis of nanoparticles from polymerization of monomers, emulsion and suspension polymerizations are of great importance in industrial applications as they do not harm the environment, it is easy to remove the heat of reaction during polymerization, and they ensure feasible handling of the final product with low viscosity [51,52] (Figure 2). Table 1 presents the methods of synthesis and characteristics of the nanoparticles studied in the hybrid systems reported.

The current limitations of nanoparticles include their high operational cost, limited suitability (for laboratory use only), and expertise requirement. The characteristics of polymeric materials used to develop nanoparticles, such as the acidity of poly(D,L-lactide-co-glycolide) (PLGA), make it less suitable for carrying drugs and bioactive molecules [53].

Some polymers chosen as raw material for the encapsulation of bioactive compounds are poly(ethylene oxide) (PEO), poly(vinyl alcohol) (PVA), poly(caprolactone) (PCL), and poly(vinyl pyrrolidone) (PVP), which have low toxicity.

Encapsulated compounds are of interest to the food industry, as well as to the pharmaceutical industry, such as the polyphenols. Their antioxidant activities make them attractive to be encapsulated to maintain their bioactivity [53,54].

Substances that are prime candidates for encapsulation are unstable compounds, as they tend to oxidize easily, especially if they are exposed to light, heat, or oxygen. Hydrophobic compounds are also candidates for encapsulation since they have some limitations regarding integration with formulations of either food or drug delivery systems. Encapsulation with the electrospinning technique seeks to protect substances from an aggressive environment, to be released over a long period or at specific sites [18,27].

Another use of encapsulation is in the administration of different drugs at the same time and in the same place.

There are different techniques for the encapsulation of bioactive molecules, such as spray drying, phase separation, emulsion processes, and direct blend electrospinning. They all have different drawbacks. To overcome these barriers, researchers have developed nanofibers for encapsulation made from coaxial electrospinning tailored to drug specificities and delivery system requirements [55]. This is the case with the dual drug delivery system for tissue regeneration used by Mohamady Hussein et al. They developed core-shell type nanofibers using coaxial electrospinning, incorporating phenytoin into the shell, and using the drug as a proliferative agent to help cell regeneration. PCL was chosen for the shell material due to the hydrophobic nature of the drug, with the intention of controlled release and improved mechanical properties. A hydrophilic core composed of PVA was selected for the nanofibers. Since the humidity of the microenvironment is proposed to increase the efficiency of wound healing, silver nanoparticles with chitosan were incorporated as biocidal agents [56,57]. Chitosan also has the quality of promoting the formation of collagen when it breaks down in contact with wounds, which also helps wound healing [58]. The latter is an innovative scaffold system with multiple applications and specific functions.

**Table 1 micromachines-14-00208-t001:** Summary of the nanoparticles in hybrid systems. Applications, synthesis, size, and loading method used.

Nanoparticles	Application (Advantage)	Synthesis Method	Size Diameter Nanoparticles	Loading Method and Drug Loaded	References
Bacterial cellulose whisker (BCW NP)	Increase the filler density in composite nanofibers	Plasma treatment using a microwave oven	Nanoparticles of an average size of 49.1 ± 13 nm	Without drug	[59]
functionalized bacterial cellulose whisker (f-BCW NP)	Increase the filler density in composite nanofibers. Increased the glass transition temperature	Plasma treatment using a microwave oven	Nanoparticles of an average size of 49.1 ± 13 nm	Without drug	[59]
L-lysine based nanogel	Increased in varying degrees. The response to ammonia	In vitro enzymatic biodegradation of 4-Lys-macrogel	Nanogels size was about 60 nm with a narrow size distribution	Without drug	[33]
Chitosan (CS)	Antibacterial activity	Oil-in-water (o/w) type emulsion, ionic gelation	Average particle size of the five nanoparticles ranged from 94.3 ± 2.1 to 246.1 ± 6.3 nm with the in- crease of chitosan concentration.	Oil-in-water (o/w) type emulsion. Moringa oil	[60]
Chitosan	Improvement in the morphology, thermal, mechanical, antibacterial properties and cytobiocompatibiliy Biocidal agents to potentiate the wound healing process	Nanoparticles thermally synthesized with chitosan acting as both a capping and reducing agent	Average particle size of 53.6 ± 20.5 nm	Thermally synthesized Silver	[56]
Chitosan	Wound dressing, wound healing with antibacterial, antioxidant, and cell proliferation properties	Electrosprayed	Curcumin (CUR) loaded into chitosan nanoparticles average diameter: 32.17 nm	Nanoencapsulation of curcumin by emulsified curcumin-chitosan solution	[57]
Poly(N-isopropylacrylamide-acrylic acid)	Promote the wound healing process to achieve higher wound healing efficacy	Free radical nanoprecipitation polymerization	Average size ~100 nm	Without drug	[61]
Lignin	Cell viability and differentiation, along with neurite length extension were promoted	Commercially available	Average size ~90 nm	Without drug	[62]
Lignin	Bone tissue engineering	Precipitation, via a simultaneous pH and solvent shifting technology	Average size count at 100–200 nm range	Without drug	[63]
Keratin	Potential neural tissue applications	Electrospray deposition	Average size 250–350 nm range	Without drug	[64]
Carboxymethyl-hexanoyl/chitosan	Increased its cytotoxic effect in melanoma cells	Ionotropic gelation	Average size of 32.6 ± 1.2 nm	Pyrazoline. In situ	[65]
Chitosan-aniline nanogels	Bactericide	Complexation-reduction	Average size of 78 ± 19 nm	Silver nanoparticles In situ	[66]
Chitosan-tripolyphosphate	Encapsulation and release of therapeutic proteins	Ionotropic gelation	Average size 194 ± 3 nm	Transforming Growth Factor β3 (TGF-β3) In situ	[67]
Carboxymethyl chitosan	Exhibits antibacterial properties and promotes skin wound healing	Electrostatic droplet	Average size 164.6 ± 5 nm	Encapsulates antibacterial peptide (OH-CATH30). In situ	[23]
Chitosan	Vaginal controlled release of benzydamine	Ionic gelation	Average size 128–710 nm range. (Depending on the formulation)	Benzydamine. In situ	[68]
Chitosan	Antibacterial. Encapsulation of antibacterial agent	Ionic gelation	Average size 10–25 nm range	Antibiotics (e.g., ciprofloxacin, ofloxacin, levoxacin, gemifloxacin). Absorption	[69]
Poly (methacrylic acid)	Peripheral nerve regeneration.	Suspension polymerization. Molecular imprinted	Average size of 80–115 nm range	4-aminopyridine (4-AP). Entrapping the 4-AP molecules via hydrogen bond formation.	[54]
Poly (lactic-co-glycolide)	Topical vaginal drug delivery	Nanoprecipitation. Passive PEGylation with Pluronic^®^	Average size 172 ± 19 nm	Rhodamine, directly conjugated. Antiretroviral drug etravirine (ETR). In situ	[70]
Chitosan	Antibacterial, mucoadhesive, encapsulation of drug	Ionic gelation	Average size ~300–400 nm	Epinephrine. In situ	[71]

## 4. Incorporation of Nanoparticles into Nanofibers

The first nanofiber systems with polymeric nanoparticles mentioned in the literature was reported by Yoon, who studied hybrid systems of bacterial cellulose nanoparticles (BCW NPs) and functionalized bacterial cellulose nanoparticles (f-BCWNPs) in poly(ethylene oxide (PEO) nanofibers. TEM analysis demonstrated the embedding of nanoparticles in the nanofibers, but there was agglomeration and deformation of the embedded nanoparticles [59].

For the incorporation of nanoparticles in nanofibers, three main techniques were identified. The most commonly mentioned one is direct blending electrospinning, where the nanoparticles are encapsulated and/or entrapped in the nanofibers. For this purpose, the nanoparticles are incorporated by mixing them in the solution together with the base polymer solution before being electrospun (Figure 3a). Another way to encapsulate nanoparticles is through coaxial electrospinning, in which core-shell type nanofibers are created. These follow the same principle as the previous ones; the nanoparticles are mixed in the solution before being electrospun, but the difference is that two solutions are spun simultaneously using a needle with two different capillaries [72]. Another approach is to attach the nanoparticles to the surface of nanofiber mats by directly depositing them on previously prepared nanofiber mats by means of the electrospray technique (Figure 3b). The third option is to impregnate the nanoparticles in the preformed nanofibers’ mats by immersing the scaffolds in solutions containing nanoparticles (Figure 3c) [71,73].

## 5. Hybrid Systems for Control of Reactive Oxygen Species for Tissue Regeneration

The enzymes responsible for maintaining a balanced and adequate homeostasis in the skin are particular, providing focused mechanisms for redox mediation in homeostasis. This redox mediation system involves some antioxidants of organic origin and enzymes such as glutathione, peroxidases, catalase, superoxide dismutase, and thioredoxins [74]. Homeostasis is an essential process that keeps cell health optimal. Reactive oxygen species (ROS), such as hydroxyl radicals, singlet oxygen, peroxides, and superoxide, are actively involved in biochemical processes such as angiogenesis, collagen synthesis, epithelialization, and body protection against infectious pathogens [75].

In damaged tissue, a series of inflammatory processes occur that lead to tissue repair, but if the tissue is pathological, such as in cases of immunosuppression, diabetes, or advanced age, the proinflammatory cytokines generated by immune cells do not generate ROS appropriately. Since ROS also participates in the translation of signals in response to cellular oxidative stress, they can generate an imbalance in the cellular microenvironment. If ROS levels are inadequate, this can be counterproductive for tissue healing. Excessive ROS for prolonged periods is also harmful and generates signals that lead to cell apoptosis. One alternative to improving the regeneration of damaged tissue is to control ROS levels through biomaterials and nanotechnology [76,77,78].

Zhang et al., developed a hybrid system made of nanofibers with nanogels evenly distributed in the membrane. The system is fabricated using the airbrush technique with poly (L-lactic acid) (PLLA) nanofibers and poly (N-isopropylacrylamide-acrylic acid) (PNA) redox-sensitive nanogels that act as a reservoir of disulfide bonds for the mediation of redox potential to promote the wound healing process. They demonstrated that the incorporation of nanogels helped the healing of skin wounds, both in healthy tissue and diabetic tissue [61].

The excessive generation of ROS is detrimental to the regeneration of nervous tissue; it has been shown that materials such as PCL induce an inflammatory response that generates too much ROS, causing tissue damage. To solve this problem, it is necessary that the material used in the nanofibers have antioxidant properties, although PCL showed inferior antioxidant activity [79]. Nerve cells would create oxidases of nicotinamide adenine dinucleotide phosphate and ROS, resulting in DNA necrosis. Nervous tissue, involved in a microenvironment with H_2_O_2_, increases the amounts of ROS intracellularly, which increases oxidative stress and apoptosis. The oxidation diffusion reaction can end with some functional groups, such as methoxyl and hydroxyl, that provide hydrogen, inhibiting cell death by increasing ROS [80].

Some of the research focused on tissue engineering and wound healing have incorporated lignin into their studies due to its antioxidant capacity. This is because it scavenges oxygen free radicals and can stabilize the reactions initiated by these radicals [81]. With this, the issue of low antioxidant capacity in PCL nanofiber systems can be overcome. The project by Amini et al., focused on the regeneration of nervous tissue. They found in vivo and in vitro that PCL nanofibers with lignin nanoparticles in different concentrations promoted viability and cell differentiation by increasing the percentage of lignin nanoparticles in the PCL nanofibers. The expression of neuronal markers for differentiated cells was also increased with a higher percentage of lignin nanoparticles incorporated into the PCL nanofibers. Regarding the in vivo groups that showed better regeneration, 15% lignin nanoparticles were incorporated, demonstrating that their hybrid nanofiber/nanoparticle system has excellent potential to be used in the cellular regeneration of nervous tissue [62]. The potential of hybrid systems of PCL/lignin nanofibers and nanoparticles, respectively, is not only reduced to the regeneration of nervous tissue. In the work by Haider et al., the properties of lignin were exploited, synthesizing nanoparticles to take advantage of its antimicrobial qualities since the oxygen contained in the functional groups of its phenolic parts (–OH, –CO, COOH) present resistance against multiple microorganisms [82]. They developed PCL nanofiber scaffolds with lignin nanoparticles and observed a significant improvement in osteogenic cell differentiation versus pure PCL scaffolds. They integrated the antibacterial and osteostimulative capacities of nanoparticles with the mechanical properties of PCL nanofibers, demonstrating the potential of this type of hybrid nanodevice for bone tissue engineering [63].

## 6. Materials Used in Hybrid Systems

Several materials have been used as base materials to form the nanofiber mats and incorporate nanoparticles for diverse applications. Findings in the literature are summarized in Table 2 and surveyed in the present section. The structure of the polymers is presented in Figure 4.

### 6.1. Poly(D,L-Lactide-co-Glycolide) (PLGA)

PLGA nanofibers made by the electrospinning method have a morphology that contributes to nanoparticle retention. PLGA nanofiber plates have been produced to improve efficacy and safety in extended-release systems for analgesics, especially for local analgesia [83]. Another way in which nanofibers can be used to improve the bioavailability of a drug is in gastroretentive delivery systems. This is due to properties such as mucoadhesion, which provides longer exposure time of the drug within the intestinal tract [84]. The goals of an effective delivery system include the delivery of a specific amount of drug to a target site for a defined period. Despite the many advantages of nanofiber mats created by electrospinning, in a delivery system based on nanofibers, the drug mixed or encapsulated in the nanofiber matrix has one drawback when it comes to delivery: the “burst release” of the drug in the pharmacokinetic profile. This occurs when the drug is weakly anchored to the large surface of the polymeric mat. This problem can be solved by encapsulating the drugs within nanoparticles and then loading the nanoparticles into electrospun mats [85,86].

### 6.2. Keratin/Poly (Vinyl Alcohol) (PVA)

Biocompatibility is another highly desirable characteristic in delivery systems. Keratin combined with polymeric nanofibers is one option that has been investigated for this purpose. Another option involves coating nanofibers with keratose, a biomaterial that is extracted from human hair fibers by protein oxidation [87]. Oxidized keratins or keratose nanoparticles negatively compromised the mechanical properties of keratin/PVA blend nanofibers, keratins exhibited serious deficiencies, including low viscosity and complications with the electrospinning process. A new method of keratin modification was developed: PVA nanofibers coated with keratin nanoparticles were prepared by electrospray deposition after electrospinning, resulting in a simultaneous improvement of their mechanical properties and biocompatibility [64,88]. 

### 6.3. Chitosan

The use of natural polymers (also known as polysaccharides) offers antimicrobial, antioxidant, and anti-inflammatory features that help cell proliferation and decrease infections, all of which provides an improvement in the wound healing process. They are excellent biomimetics due to their biocompatibility and null response from the immune system [89]. 

Chitin and its deacetylated form, chitosan, have biocompatible and biodegradable properties, among other desirable characteristics in the field of biomedicine. Chitosan’s characteristics include non-toxicity, antibacterial activity, the ability to form films, and the chelation of metal ions. This is attributed to the presence of free protonatable amino groups along the chitosan skeleton, which provides rigidity due to the repulsive forces between the protonated -NH_2_ groups. Unfortunately, this property limits its electrospinning capacity. For this reason, chitosan nanofibers are combined with other more flexible materials, such as PEO, to form electrospun mats through coaxial-electrospinning [90,91]. 

The rigid backbone and high molecular weights make chitosan solutions very viscous, so the electric field is stretched. To reduce surface tension, researchers use acetic acid (90%) as a concentrated solvent and a combination of solvents such as 1,1,1,3,3,3-hexafluoro-2-propanol, trifluoroacetic, and dichloromethane; these are used for the development of chitosan-based electrospun nanofibers. To crosslink the nanofibers, the authors use glutaraldehyde in steam after electrospinning, which helps preserve the fibers’ structure when the scaffolds are immersed in aqueous media. Another of the strategies used to produce soft nanofibers was to use blends of chitin or chitosan with other polymers such as PVA or PEO [92]. The change in the surface of the nanofibers, either with the immobilization or functionalization of other nanomaterials (nanogels, nanocapsules, liposomes, and nanoparticles in general), results in scaffolds with new capacities, features, properties, or functions. These intelligent nanomaterials can have photocatalytic, self-cleaning, or antimicrobial characteristics that are activated by stimuli; light absorption, for example [93]. Ballesteros et al., developed, characterized, and compared different combinations of smart nanomaterials. They used PCL nanofibers as a base, which were functionalized with a combination of photosensitive aniline-chitosan nanogels and silver nanoparticles to obtain a hybrid device with antimicrobial characteristics and the ability to control release response after being exposed to laser light [66]. As the literature indicates, chitosan nanoparticles can be formed by ionotropic gelation. Well-defined nanogels have also been reported, using tripolyphosphate as a crosslinker, with potential for the development of drug delivery systems. Combining the positive zeta potential with polyanionic biopolymers can form hydrogels by deposition of the nanogels layer by layer for a multilayer coating that dries into a homogeneous film [69]. 

Hydrogels with a porous structure that allow a large volume of water have also been reported, which provide a more favorable environment to encapsulate biological products such as cells and proteins [94]. The composite systems also provide protection against degradation [95]. 

A different system for supplying drugs was developed, protecting active peptides from proteases and properly releasing these peptides. It was reported that a ready-made biodegradable drug nanoparticle delivery system based on carboxymethyl chitosan (CMCS) can promote the sustained release of the antibacterial peptide OH-CATH30 (OH-30). The investigation results also showed that the prepared CMCS-OH-30 nanoparticles sped healing and reduced skin injury time in mice studies [23].

Fatmanur Tuğcu-Demiröz et al., developed a hybrid system of nanofibers loaded with chitosan nanoparticles for controlled vaginal release of benzydamine. They obtained the controlled release of the drug with formulations loaded with nanoparticles. Although gel formulations had better mucoadhesive properties, the nanoparticle-loaded nanofiber formulations provided greater drug penetration through vaginal tissue. The results showed that the increased surface area, high solubility, high wettability, and porosity of nanofibers containing dispersed nanoparticles were parameters important for the effectivity of the system [68]. 

Felipe et al., developed a chitosan/PEO nanofiber drug delivery scaffold system with carboxymethyl-hexanoyl chitosan-dodecyl sulfate (CHC-SDS-H3) polymeric nanoparticles loaded with H3TM04 (1-(5-(naphthalen-2-yl)-3-(3,4,5-trimethoxyphenyl)-4,5-dihydro-1H-pyrazol-1-yl) ethenone). The literature mentions that pyrazoline compounds have a wide spectrum of pharmacological activity as anti-inflammatory, antibacterial, antifungal, antipyretic, analgesic, antiviral, antioxidant, anti-angiogenic, and anti-tumor compounds. The drug is of great importance for skin cancer therapy; studies have shown that it induces apoptosis in tumor cells, which results in the reduction or elimination of the tumor. In conjunction with nanofiber systems, which exhibit similar properties to the natural fibrillar structure, hybrid systems could be used for the controlled release of drugs in the treatment of local anti-melanoma chemotherapy [65,96]. 

### 6.4. Poly(Vinyl Alcohol) (PVA)

Pour Khalili et al., directed their contributions toward supporting the reduction of risks associated with the inappropriate use of antibiotics that create bacterial resistance. They opted to work with nanofibers made of polyvinyl alcohol (PVA), which were functionalized with tripolyphosphate (TPP)/chitosan nanogels. These nanogels were loaded with the antibiotic gemifloxacin. The nanogels were electrosprayed to be immobilized on the nanofiber mat, obtaining a hybrid release system in which the drug is encapsulated with the aim of being used as an antibacterial for wound dressing with a constant and controlled release of the drug [68,69,97].

### 6.5. Poly(Vinyl Pyrrolidone) (PVP)

Materials such as polyvinylpyrrolidone are used for their biocompatibility in the electrodynamic method to create nanomaterials. This polymer can be mixed with other biomaterials, for example, PCL, in such a way that nanofibers can be developed for healing tissues, helping to dress wounds and supporting their healing [97,98].

PVP nanofibers with mucoadhesive properties can increase the interaction with the vaginal mucosa and release the drug over the required time. One of its qualities widely mentioned in the literature is mucoadhesion, which is used to overcome issues such as low drug availability in vaginal applications since its mucoadhesive characteristic provides a great advantage by increasing interaction with the vaginal mucosa and releasing the drug over the required time. Hybrid nanofiber/nanoparticle systems are considered more suitable for vaginal formulations, as they achieve high permeability through the vaginal mucosa and controlled release.

Krogstad et al., studied models of hybrid delivery systems of PVP nanofibers and PVA nanofibers incorporating PLGA nanoparticles loaded with a fluorescent compound. Their study confirms that these delivery systems have the advantage of avoiding problems in topical vaginal application, such as low retention and leakage [70]. 

### 6.6. Poly(Caprolactone) (PCL)

Poly (caprolactone) (PCL) is another synthetic polymer, and was used in the work of Sundhari et al., where created PCL nanofibers proved to be a good scaffold for the promotion of cell differentiation. The structure of these nanofibers presented a smooth and uniform diameter, and with certain wettability of the nanofibers, there is potential to facilitate neurite outgrowth, promoting cell viability by including lignin nanoparticles in the PCL nanofibers [99].

### 6.7. Poly(Vinylidene Fluoride) (PVDF)

Piezoelectric scaffolds implanted at a bone defect or injury site trigger electrical stress-induced stimulus through physiological loads. Such hyperpolarization activates calcium channels with cell membrane voltage. Intracellular Ca^2+^ ions play an essential role in cell proliferation.

Polyvinylidene fluoride (PVDF) and its composites are biocompatible piezoelectric materials that show high efficiency for bone regeneration. The beta phase is the crystalline phase of PVDF with the most piezoelectric properties. Research showed that the electrospinning method is the most efficient for fabricating the 3D porous structure of the piezoelectric b-PVDF. The porous structure formed and the polarization improves biocompatibility growth in PVDF scaffolds. Different nanoparticles have also been incorporated in PVDF scaffolds to improve osteoinductive and mechanical properties. These include antimicrobial, antioxidant, and anti-inflammatory functionalities. Among the materials studied are hydroxyapatite, graphene oxide, ZnO, and polyhedral oligomeric silsesquioxane–epigallocatechin gallate conjugate [100].

### 6.8. Polymer Mixtures

Nanofibrous scaffolds based on mixtures of poly (L-lactide-D, L-lactide)/poly (acrylic acid) [PLDLLA/PAAc] in the presence of poly (2-hydroxyethyl methacrylate) have been prepared as molecularly imprinted polymer nanoparticles to enhance osteogenesis. By adding 10% by weight of PAAc to PLDLLA and using the response surface technique, the average diameter of the electrospun nanofibers obtained was 237 nm and increased the osteogenesis yield of optimized nanofiber scaffolds. In general, by minimizing the diameter of nanofibers, their specific surface area increases. Furthermore, the minimal size, together with a uniform morphology, of the non-beaded nanofibers leads to increased cell adhesion, improved drug loading, and controlled release [54,101,102]. 

## 7. Conclusions

Hybrid systems of polymeric nanofibers and nanoparticles represent the potential to improve the characteristics required for their biomedical use. They are also of interest in the food industry since they can be prepared with biocompatible and biodegradable polymeric materials.

The combination of these nanometric-dimensioned materials (nanofibers and polymer-based nanoparticles) provides advantages for being used together as hybrid systems and enhancing or developing properties that are not present individually. These properties include increases in surface-volume ratio, drug encapsulation, protection of proteins and bioactive substances sensitive to degradation, and improved cost-benefits of their development, among others. In the case of bio-drugs, the hybrid systems exhibit unique pharmacokinetics and pharmacodynamics.

The main application of nanoparticle/nanofiber hybrid systems is the avoidance of drug “burst release,” which occurs in systems using individual technologies. Most hybrid systems studied have applications for achieving sustained release of hydrophilic drugs.

Electrospun mats have mucoadhesive properties that improve residence times at the delivery site. Wound healing and tissue engineering scaffolds help cell proliferation and contribute to an improvement in both aesthetic and functional healing. The combination of mats with drug-loaded nanoparticles helps with greater penetration into the tissue of interest, such as vaginal or buccal delivery. Some materials are also useful due to their antibacterial properties, as in the case of chitosan nanoparticles.

Techniques used for mixing or incorporating polymeric nanoparticles into polymeric nanofiber mats, such as electrospinning, electrospray, and coaxial electrospinning, are widely studied techniques. This new information provides advantages when developing this type of hybrid nanotechnology.

A feasible combination of both techniques and materials must be sought. One complication is that some polymeric materials are difficult to electrospin. In the same way, other difficulties arise in investigating these systems; for example, the change in the mechanical properties of the nanofibers, the agglomeration of nanoparticles at the time of their incorporation into the mats, and irregular distribution issues. These limitations must be overcome in order to reveal their great potential for future research.

Researchers can take advantage of the characteristics of different polymeric materials that are used in these hybrid systems and may be of interest for future research in which the most convenient combination of polymeric materials is chosen for the desired purpose.

Electrospun mats share properties with hydrogels (three-dimensional crosslinked polymeric networks), such as acting as templates, exudate-absorbents, breathable barriers, and controlling drug diffusion. Hydrogels, for instance, are widely used in biomedical applications such as tissue engineering and drug delivery. Hydrogels can be produced more quickly compared to electrospun mats. Hydrogels can also be responsive to temperature, solvent, and bioelectricity, which makes them suitable for bionic skin. Recent studies relate to the 3D printing of hydrogels to be used in robotics, with promising results. This is not the case with electrospun mats, where applications are limited to tissue engineering and drug delivery. Setup for electrospinning is simple, however, and cost-efficient compared to 3D printing. One disadvantage of hydrogels is that they are prepared by the polymerization of monomers and crosslinkers, which can leach if unreacted and cause toxicity. For the preparation of hydrogels from biopolymers, the latter must be derivatized with polymerizable moieties for crosslinking. On the other hand, electrospun mats are prepared from prefabricated biocompatible and biodegradable polymers [103,104,105,106,107].

The incorporation of nanoparticles in hydrogels produces entanglement of the nanoparticles within the hydrogel network. Nanoparticles in hydrogels improve the mechanical properties of the material, but the advantages of nanoparticles are not observed in these systems, such as avoiding burst release of loaded drugs or improving penetration into tissues [108,109,110].

One of the major drawbacks for the application of nanofibrous mats in nanomedicine is the reduced number of in vivo studies in humans [111]. We were unable to find studies of biocompatibility and/or effectivity in volunteers for nanoparticle/nanofiber systems. Most of the reported systems require further investigations before products are ready for commercialization.

## Figures and Tables

**Figure 1 micromachines-14-00208-f001:**
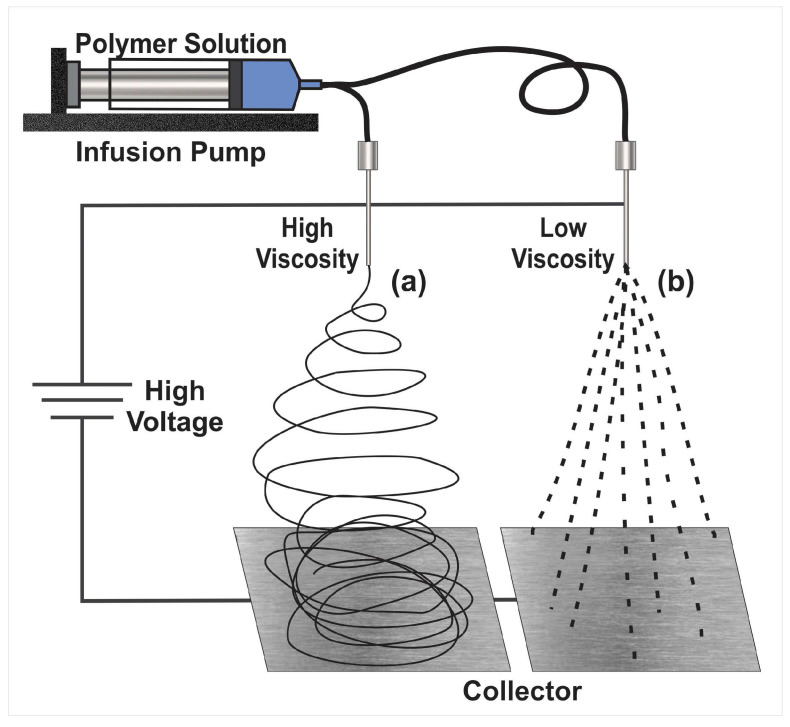
Basic setup for the electrodynamic method to produce either nanoparticles or nanofibers. (**a**) Electrospinning and (**b**) electrospray.

**Figure 2 micromachines-14-00208-f002:**
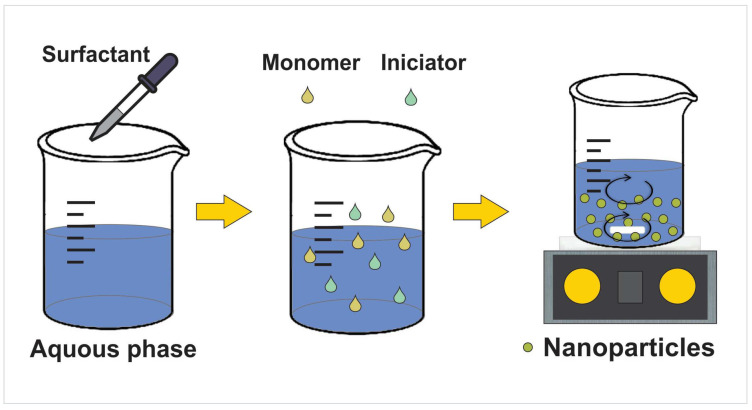
Preparation of nanoparticles by emulsion polymerization.

**Figure 3 micromachines-14-00208-f003:**
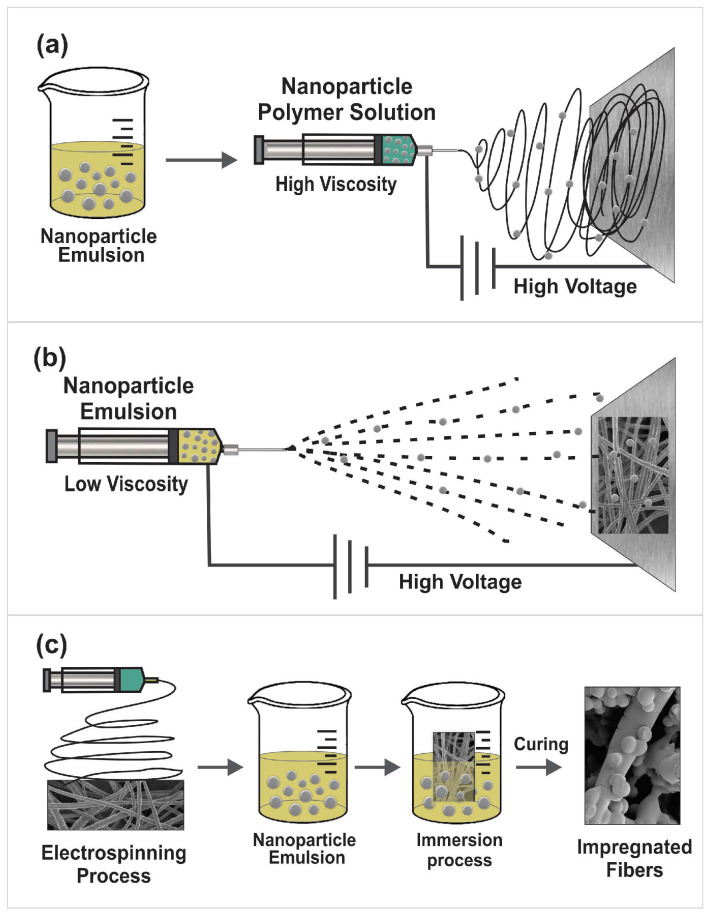
Incorporation of loaded nanoparticles into electrospun mats: (**a**) incorporation of nanoparticles by mixing in the polymer solution; (**b**) depositing nanoparticles on nanofiber mat via electrospray; and (**c**) impregnation of nanofiber mats with nanoparticles via immersion.

**Figure 4 micromachines-14-00208-f004:**
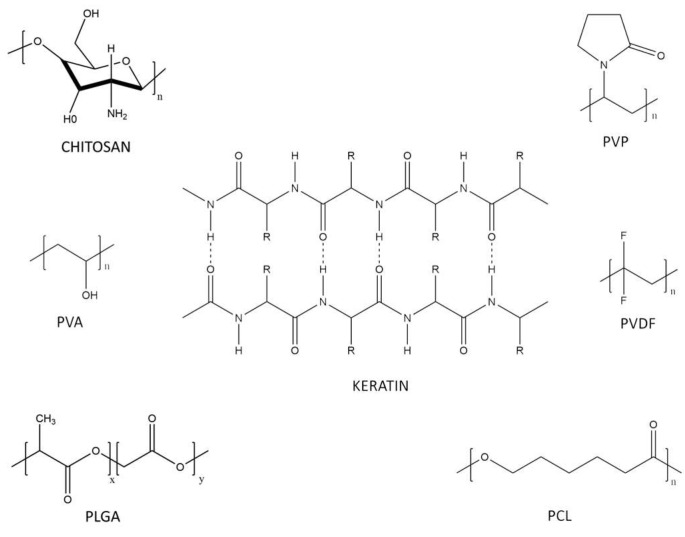
Biopolymer structures used in nanofibers and nanoparticles.

**Table 2 micromachines-14-00208-t002:** Summarized organic nanoparticles/nanofibers hybrid systems.

Nanoparticles	Nanofibers	Applications	Method	Size Diameter Nanoparticles	Size Diameter Nanofibers	References
Bacterial cellulose whisker (BCW NP) and functionalized bacterial cellulose whisker (f-BCW NP)	Poly (ethylene oxide) (PEO)	Improving the thermal characteristics of bacterial cellulose whisker for biological applications.	Electrospinning. Plasma treatment using a microwave oven.	Nanoparticles of an average size of 49.1 ± 13 nm, concentrations of nanoparticles (0.05, 0.1, 0.3, and 0.5 wt%)	The diameter of PEO nanofibers was 0.37 ± 0.09 μm. The diameter of PEO/BCW NP and PEO/f-BCW NP composite nanofibers was about 0.38 ± 0.12 μm and 0.46 ± 0.08 (0.05%) 0.41 ± 0.21 and 0.57 ± 0.07 μm (0.1%). Showed heterogeneous nanostructures with various sizes and sometimes bead-like structure (0.3% y 0.5%)	[59]
L-lysine based nanogel	Poly (aniline)/poly (acrylonitrile) (PANI/PAN)	Ammonia sensor based	Electrospinning. Enzyme biodegradation	Nanogels size was about 60 nm with a narrow size distribution	The diameters of the fibers PANI/PAN 250–300 average	[33]
Chitosan (CS)	Gelatin (type B from porcine skin)	Biocontrol of *Listeria monocytogenes* and *Staphylococcus aureus* on cheese	Electrospinning. Emulsion and ionic gelation	Average particle size of the five nanoparticles ranged from 94.3 ± 2.1 to 246.1 ± 6.3 nm with the increase of chitosan concentration	The thickness of the gelatin nanofibers was 0.102 ± 0.004 mm	[60]
Chitosan	Core-shell PVA/PCL	Cell proliferation and antibacterial activity for tissue regeneration and wound healing.	Coaxial electrospinning. Nanoparticles Thermally synthesized with chitosan acting as both a capping and reducing agent	Average particle size of 53.6 ± 20.5 nm	The diameters of the fibers 70 nm average	[57]
Chitosan	(PCL)/Chitosan	Wound dressing, wound healing with antibacterial, antioxidant, and cell proliferation properties.	Electrospinning. Electrosprayed	Chitosan nanoparticles average diameter: 25.46 nm and curcumin (CUR) loaded into chitosan nanoparticles average diameter: 32.17 nm.	PCL nanofiber (mean diameter 214.9 nm) PCL/CS nanofiber (mean diameter 115.6 nm). PCL/CS/CUR nanofiber (mean diameter 100.08 nm). PCL/CS/CUR electrosprayed with CURCSNPs (mean diameter 99.84 nm).	[57]
PNA	PLLA	Promote the wound healing process to achieve higher wound healing efficacy	Airbrushing. Radical emulsion polymerization	Average size ~100 nm	PLLA nanofiber diameters of 100–500 nm	[61]
Lignin	PCL	Potential to be applied for nerve regeneration	Electrospinning. Kraft lignin nanoparticles (Mw = 9000; Sigma-Aldrich; Germany)	Average size ~90 nm	The mean diameter of 0, 5, 10 and 15% Lignin fibers was 269 ± 60, 291 ± 82, 334 ± 86, and 364 ± 97 nm, respectively	[62]
Lignin	PCL	Bone tissue engineering	Electrospinning. Simultaneous pH and solvent shifting	Average size count at 100–200 nm range	Diameter of the nanofibers, ranging from 400 to 2200 nm	[63]
Keratin	PVA	Potential neural tissue applications	electrospinning and electrospray deposition	Average size 250–350 nm range	No information	[64]
Carboxymethyl-hexanoyl chitosan	PEO-chitosan	Skin cancer treatment	Electrospinning. Ionotropic Gelation	Average size of 32.6 ± 1.2 nm	The average size of the PEO-CS nanofibers was 157.1 ± 5.0 nm. With nanoparticles 197.8 ± 4.1 nm)	[65]
Silver nanoparticles, Chitosan-aniline nanogels	PCL	Antibacterial Applications	Electrospinning complexation-reduction, ionotropic Gelation	Average size of 78 ± 19 nm	The nanofibers presented diameters of 240 ± 70 nm	[66]
Chitosan	PCL	Drug delivery system: encapsulation and release of therapeutic proteins (Transforming Growth Factor β3)	Electrospinning. Ionotropic Gelation	Average size 194 ± 3 nm	The layer thickness of the multilayer systems in the dry state was determined as a function of the number of layers	[67]
carboxymethyl-chitosan	PVA/chitosan	Encapsulates antibacterial peptide (OH-CATH30) Exhibits antibacterial properties and promotes skin wound healing	Electrospinning, electrostatic droplet	Average size 164.6 ± 5 nm	The mean diameter of the control group (mean ± SD) was 357.34 ± 110.45 nm, which de- creased from 371.80 ± 110.31 nm to 327.48 ± 114.28 nm after the drug was loaded	[23]
Chitosan	Poly (vinylpyrrolidone) (PVP)	Vaginal controlled release of benzydamine	Electrospinning. Ionic gelation	Average size 128–710 nm range. (Depends of the formulation)	Average size 436 ± 155 nm and 557 ± 221 nm	[68]
Chitosan	PVP	Potential application as antibacterial fabrics for wound dressings. Mats are especially applicable for the treatment of diabetic wounds	Electrospinning. Ionic gelation	Average size 10–25 nm range	Average size 150 nm to 250 nm range	[69]
Methacrylic acid	Poly (L-lactide-co-D, L- lactide) containing multi-walled carbon nanotubes	Peripheral nerve regeneration	Electrospinning, Emulsion polymerization. Molecular imprinted	Average size of 80–115 nm range	Average size 92 nm	[54]
Poly lactic-co-glycolide	PVP, PVA	Topical Vaginal Delivery Platform for nanoparticles	Electrospinnning nanoprecipitation	Average size 172 ± 19 nm	The diameter of the composite fiber and the diameter of the nanoparticles were around 200–300 nm, with PVA fibers having a mean diameter of 248 ± 88 nm, PVP fibers of 297 ± 125 nm	[70]
Chitosan	Gelatin	For rapid hemostasis	Electrospinning. Ionic gelation	Size of ~300–400 nm	The diameter of the nanofibers is about 305 ± 50 nm	[71]

## Data Availability

Not applicable.

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
