# Peer review of "Hybrid Systems of Nanofibers and Polymeric Nanoparticles for Biological Application and Delivery Systems"

_micromachines, 2023, doi:10.3390/mi14010208_

Round 1
Reviewer 1 Report
This review summarizes the hybrid systems of polymeric nanoparticles and polymeric nanofibers, critically analyzing how the combination improves the properties of the materials and contributes to the reduction of some disadvantages found in nanometric devices and systems. Some issues should be addressed before any further consideration:
-the keywords don’t reflect the goal of the review paper, please replace them with better keywords
-I suggest adding electrospinning to the keywords
-it is suggested to add the following papers and discuss them properly to enrich your review paper: https://doi.org/10.1016/j.apsusc.2022.155290, https://doi.org/10.2147/IJN.S243223, https://doi.org/10.1080/09205063.2022.2065409
-it is better to summarize section 3: nanoparticle in a Table
-please add the structure of polymers in section 7: Materials used in hybrid systems
-what about the challenges of the hybrid system in medicine, the authors didn’t discuss this matter very well
-Please add a graphical abstract to your paper that presents the whole manuscript
Author Response
Reviewer 1
This review summarizes the hybrid systems of polymeric nanoparticles and polymeric nanofibers, critically analyzing how the combination improves the properties of the materials and contributes to the reduction of some disadvantages found in nanometric devices and systems. Some issues should be addressed before any further consideration:
R1.1.- The keywords don’t reflect the goal of the review paper, please replace them with better keywords
Response: Thank you for your comments. Done
Location: Keywords
R1.2.- I suggest adding electrospinning to the keywords
Response: Included
Location: Keywords
R1.3.- It is suggested to add the following papers and discuss them properly to enrich your review paper: https://doi.org/10.1016/j.apsusc.2022.155290, https://doi.org/10.2147/IJN.S243223, https://doi.org/10.1080/09205063.2022.2065409
Response: The papers were added and discussed
Location: 1. introduction, 2.1 Electrosspining, 6.6. Polyvinydiene fluoride, References [30], [1], [103]
R1.4.- It is better to summarize section 3: nanoparticle in a Table
Response: Table 1 was introduced to summarize section 3
Location: 3. Nanoparticles’ preparation and drug loading
R1.5.- Please add the structure of polymers in section 7: Materials used in hybrid systems
Response: Figure 4 with the structure of the polymers in section 7 (now section 6) was introduced
Location: Section 6.
R1.6.- What about the challenges of the hybrid system in medicine, the authors didn’t discuss this matter very well
Response: Up to now there are only a few studies on animal model. No reports in human have been reported. This is discussed in the conclusions.
Location: 7. Conclusions
R1.7.- Please add a graphical abstract to your paper that presents the whole manuscript
Response: A graphical abstract is now included.
Location: Graphical abstract

Reviewer 2 Report
In this manuscript, the author summarized the recent development and progress of the hybrid systems of polymeric nanoparticles and polymeric nanofibers for biological application and delivery platforms. The author focused on the method for enhanced overall performances of materials. I suggest the publication of this manuscript in Micromachines, a high-performance journal if the author can address the following concerns appropriately.
1. There were too many paragraphs in the introduction part, and the author should shorten them. Generally, there were three or four paragraphs in this section. Besides, too many paragraphs also made the logic not so clear. The author should emphasize the novelty and innovation of this review.
2. The logic of this manuscript was confusing. For example, the author summarized the electrohydrodynamic method, nanoparticles, encapsulation of bioactive molecules, and incorporation of nanoparticles into nanofibers in sections 2 to 5. The reviewer did not understand the reasons for this kind of arrangement. Besides, the author emphasized the significance of the electrospinning method in the abstract, which did not appear in any subheadings. The author should re-think about this.
3. The author illustrated many kinds of biocompatible materials for biological and delivery applications. Likewise, hydrogels composed of three-dimensional (3D) crosslinked polymeric networks were also demonstrated promising for the development of the biomedical and wearable field (DOI: 10.1088/1361-6463/aa84a3; DOI: 10.1016/j.apsusc.2022.153803; DOI: 10.1002/adfm.202107437; DOI: 10.1021/acsami.2c14907; DOI: 10.1063/5.0083278; DOI: 10.1016/j.bioactmat.2022.08.017). More importantly, hydrogels have been demonstrated to be widely utilized in tissue engineering, biomedical, therapy, and drug delivery applications. The author can compare the advantage and disadvantages of hydrogels with nanoparticles or nanofibers for biomedical applications.
4. There should be many published papers regarding nanofibers and nanoparticles for biomedical applications. What is the innovation of this manuscript?

Author Response
Reviewer 2
In this manuscript, the author summarized the recent development and progress of the hybrid systems of polymeric nanoparticles and polymeric nanofibers for biological application and delivery platforms. The author focused on the method for enhanced overall performances of materials. I suggest the publication of this manuscript in Micromachines, a high-performance journal if the author can address the following concerns appropriately.
R2.1.- There were too many paragraphs in the introduction part, and the author should shorten them. Generally, there were three or four paragraphs in this section. Besides, too many paragraphs also made the logic not so clear. The author should emphasize the novelty and innovation of this review.
Response: The introduction has been resumed. The novelty and innovation of this review has been emphasized.
Location: 1. Introduction.
R2.2. The logic of this manuscript was confusing. For example, the author summarized the electrohydrodynamic method, nanoparticles, encapsulation of bioactive molecules, and incorporation of nanoparticles into nanofibers in sections 2 to 5. The reviewer did not understand the reasons for this kind of arrangement. Besides, the author emphasized the significance of the electrospinning method in the abstract, which did not appear in any subheadings. The author should re-think about this.
Response: The introduction emphasizes that the review is about the polymeric nanoparticles-polymeric fibers hybrid system. It also indicates that the manuscript overviews organic nanoparticles preparation and drug loading (condensed in one section), as well as an overview of electrodynamic method, a technique to prepare nanofibers and nanoparticles. This is to give the readers an introduction to the techniques needed to prepare the parts that are assembled in the hybrid systems analyzed. Information about electrospinning has been separated in a subheading.
Location: Sections 2, 3, 4.
R2.3.- The author illustrated many kinds of biocompatible materials for biological and delivery applications. Likewise, hydrogels composed of three-dimensional (3D) crosslinked polymeric networks were also demonstrated promising for the development of the biomedical and wearable field (DOI: 10.1088/1361-6463/aa84a3; DOI: 10.1016/j.apsusc.2022.153803; DOI: 10.1002/adfm.202107437; DOI: 10.1021/acsami.2c14907; DOI: 10.1063/5.0083278; DOI: 10.1016/j.bioactmat.2022.08.017). More importantly, hydrogels have been demonstrated to be widely utilized in tissue engineering, biomedical, therapy, and drug delivery applications. The author can compare the advantage and disadvantages of hydrogels with nanoparticles or nanofibers for biomedical applications.
Response: Advantages and disadvantages of hydrogels vs polymeric nanoparticles/nanofibers hybrid systems is included.
Location: 7. Conclusions, References [106-114].
R2.4.- There should be many published papers regarding nanofibers and nanoparticles for biomedical applications. What is the innovation of this manuscript?
Response: We completely agree with the reviewer. There are very insightful reviews in the literature regarding nanofibers, and also regarding nanoparticles. Also, there are very relevant reviews regarding inorganic nanoparticles incorporated in nanofibers (nano-in-nano), mainly for antimicrobial applications. However, we did not find a review about organic nanoparticles/organic nanofibers. This review focuses the reader to the main techniques used to prepare and drug-load organic nanoparticles, as well as about the electrodynamic technique to prepare nanofibers and nanoparticle. The manuscript mainly gives the reader an overview of the advantages of hybrid systems in nanomedicine, particularly protecting the loaded drug, avoiding the “burst effect”, and achieving better penetration of active substances in diverse application’s zones.
Location: Manuscript.

Round 2
Reviewer 1 Report
Accept in present form
Reviewer 2 Report
I suggest the publication of this manuscript.